# Molecular Characterization of CF33 Canine Cell Line and Evaluation of Its Ability to Respond against Infective Stressors in Sight of Anticancer Approaches

**DOI:** 10.3390/vetsci9100543

**Published:** 2022-10-02

**Authors:** Elisabetta Razzuoli, Chiara Grazia De Ciucis, Barbara Chirullo, Katia Varello, Roberto Zoccola, Lisa Guardone, Paola Petrucci, Danja Rubini, Elena Bozzetta, Maria Goria, Floriana Fruscione, Paola Modesto

**Affiliations:** 1National Reference Center of Veterinary and Comparative Oncology (CEROVEC), Istituto Zooprofilattico Sperimentale del Piemonte, Liguria e Valle D’Aosta, Piazza Borgo Pila 39/24, 16129 Genoa, Italy; 2Unit of Emerging Zoonoses, Department of Food Safety, Nutrition and Veterinary Public Health Istituto Superiore di Sanità, Viale Regina Elena 299, 00161 Rome, Italy; 3Istituto Zooprofilattico Sperimentale del Piemonte, Liguria e Valle D’Aosta, Via Bologna 148, 10154 Turin, Italy

**Keywords:** CF33, canine mammary tumor, gene expression, innate immunity, *Salmonella Typhimurium*

## Abstract

**Simple Summary:**

Canine mammary cancer is very common and has many similarities with human breast cancer. Risk factors, physiological and pathological behaviors, and the clinical course in dogs are very similar to humans. Several molecular similarities have also been reported, such as overexpression of EGF, proliferation markers, metalloproteinase and cyclooxygenase, TP53 mutations, and CXCR4/SDF1 axis activation. These common characteristics make these breast tumors resistant to conventional therapies. It is therefore necessary to study therapeutic alternatives. Cell lines could be helpful to test in vitro immunomodulant anti-cancer therapies, allowing a reduction of laboratory animals’ involvement in the preliminary tests and achieving results in a shorter time. Although the canine mammary carcinoma cell line CF33 has been widely used in many studies on dog mammary cancer, characterization of its gene expression profile and of the influence of infective stressors of this cell line is poor. Our study shows the interaction of CF33 and *Salmonella Typhimurium (ST)* as an infective stressor, indicating that these cells may represent an in vitro model for assessing novel therapeutic approaches using bacteria.

**Abstract:**

Spontaneous mammary tumors are the most frequent neoplasms in bitches and show similarities with human breast cancer in risk factors, clinical course, and histopathology. The poor prognosis of some cancer subtypes, both in human and dog, demands more effective therapeutic approaches. A possible strategy is the new anticancer therapy based on immune response modulation through bacteria or their derivatives on canine mammary carcinoma cell lines. The aim of the present study was to analyze the CF33 cell line in terms of basal expression of immune innate genes, CXCR4 expression, and interaction with infectious stressors. Our results highlight that CF33 maintains gene expression parameters typical of mammary cancer, and provides the basal gene expression of CF33, which is characterized by overexpression of *CXCR4*, *CD44*, *RAD51*, *LY96*, and a non-continuous expression of *TP53* and *PTEN*. No mutations appeared in the CXCR4 gene until the 58th passage; this may represent important information for studying the CXCR4 pathway as a therapeutic target. Moreover, the CF33 cell line was shown to be able to interact with *Salmonella Typhimurium (ST)* (an infective stressor), indicating that these cells could be used as an in vitro model for developing innovative therapeutic approaches involving bacteria.

## 1. Introduction

Spontaneous canine mammary tumors are the most common neoplasms in female dogs, with an incidence rate of 182 cases every 100,000 dogs per year [1,2]. In dogs, mammary intraepithelial lesions show histopathological similarities with human breast cancer. Several molecular similarities have also been reported, including overexpression of steroid receptors, epidermal growth factor (EGF), proliferation markers, metalloproteinase, cell-derived stromal factor1 (SDF1) axis activation, TP53 mutations, and cyclooxygenase and C-X-C motif chemokine receptor 4 (CXCR4) [3,4]. CXCR4 is expressed by tumors (breast, melanoma) and by a range of immune cells. It is implicated in immune response regulation [4,5,6]. Since risk factors, physiological and pathological behaviors, as well as the clinical course, are very similar to humans [1,3,7,8,9], many authors promote dog as a suitable model for studying triple-negative breast cancer [9,10,11]. However, the prognosis related to some breast cancer subtypes (triple-negative) is poor for both humans and dogs and the therapies most often used currently—surgery in association with radiotherapy and chemotherapy—are often associated with tumor relapse [7,12]. Therefore, other therapeutic approaches are strongly needed for this type of aggressive tumor.

A promising strategy could be the development of new anticancer therapies based on immune response modulation using bacteria or their derivatives [13,14,15]. Since the nineteenth century, bacteria have been used to clinically treat tumors, due to their ability to penetrate hypoxic tissue, such as the tumor environment [16,17]. However, further studies are needed before proceeding to in vivo testing. In this respect, the use of cell lines could be helpful to test in vitro immunomodulant anti-cancer therapies, allowing a reduction in the use of laboratory animals in the preliminary tests and the achievement of results in shorter time. Despite canine mammary carcinoma cell line CF33 being widely used in studies on dog mammary cancer [18,19], its basal gene expression, as well as the influence of infective stressors, are still poorly characterized, and data related to innate immune response modulation could be misinterpreted.

The aim of our study was the characterization of CF33 cells with regard to: (i) basal expression level of genes involved in the regulation of the cell cycle and innate immune response; (ii) expression of the CXCR4 exclusive receptor for SDF-1; and (iii) evaluation of the interaction of this cell line with wildtype strains of *Salmonella* spp. as an infectious stressor, whose attenuated form has already been proposed as an innovative anticancer therapeutic approach.

## 2. Materials and Methods

### 2.1. Cell Culture

CF33 cell line (canine tumor, mammary gland, IZSLER biobank OIE codex BS TCL 225) was purchased from OIE biobank at IZSLER (http://www.ibvr.org/, accessed on 26 September 2022) after 35 passages. Cells were cultured in Eagle’s Minimum Essential Medium in Earle’s (BME) supplemented with 10% (*v*/*v*) fetal bovine serum (FBS), 4 mM L-glutamine and 1% (*v*/*v*) penicillin/streptomycin. Cells were subcultured to reach three different passages (37th, 39th, and 42nd) and afterwards cryopreserved in 90% FBS and 10% DMSO (Sigma, Saint Louis, MO, USA) until use. The cells’ morphology was microscopically verified after every passage.

For all experiments, the cells were cultured into 12-well tissue plates (3 × 10^5^ cells/mL, 2 mL per well) and kept in 5% CO_2_ at 37 °C until confluence (28–32 h). Basal gene expression was evaluated after the 37th, 39th, and 42nd passages. All experiments were replicated six times.

### 2.2. Selection of Reference Genes

To choose the reference gene (RG) for normalizing gene expression data in the CF33 cell line, a panel of candidate genes in use in our laboratory was tested as described in [20]; all primers are listed in Table 1: *Ribosomal protein S5* (*RPS5*), *β 2 Microglobulin* (*B2M*), *β Glucuronidase* (*GUSB*), *Glyceraldehyde-3-Phosphate Dehydrogenase* (*GAPDH*), *Hypoxanthine Phosphoribosyltransferase 1* (*HPRT1*), *Heterogeneous Nuclear Ribonucleoprotein H (HRNPH1)*, *β Actin (BACT)*, *Signal Recognition Particle Receptor (SRPR)*, *TATA-Box Binding Protein (TBP)*, *Ribosomal Protein L13A* (*RPL13A*), *Ribosomal protein S19* (*RPS19*) and *Succinate Dehydrogenase Complex Flavoprotein Subunit A* (*SDHA*).

Extraction of total RNA was carried out from 1 × 10^6^ cells with the RNeasy Mini Kit using the Qiacube System (both Qiagen s.r.l., Milan, Italy). RNA qualitative and quantitative analysis was assessed with a Qubit 3.0 Fluorometer (ThermoFisher Scientific, Milan, Italy) and BioPhotometer (Eppendorf, Milan, Italy). RNA was retro-transcribed using a OneScript^®^ cDNASyntesis Kit (Applied Biological Materials Inc., Richmond, BC, Canada), random primers, and 50 μg of RNA. A sample without template (NTC) and another one with RNA extract without retro-transcriptase (NRT) were run as negative controls. A real-time qPCR was carried out in a CFX96™ Real-Time PCR Detection System (Bio-Rad, Milan, Italy); data analysis and RGs selection were carried out as previously described in [20]. The NormFinder algorithm (version 0.953, Andersen, Ledet-Jensen, Ørntoft, Aarhus, Denmark) was used for validating the RGs [20].

### 2.3. Basal Gene Expression Profiles

Basal gene expression of interleukins (IL) (*IL1B*, *IL2*, *IL4*, *IL5*, *IL6*, *IL10*, *IL12B*, *IL15*, *IL16*, *IL17A*, *IL18*, *IL23A*, *IL27*), *C-X-C chemokine ligand type 8* (*CXCL8*), *interferon γ* (*IFNG*), *tumor necrosis factor α* (*TNFA*), *toll-like receptors 4* and *5* (*TLR4* and *TLR5*), *breast cancer type 1 susceptibility protein* (*BRCA1*) [21], *clusters of differentiation 14* and *44* (*CD14* and *CD44*), *C-X-C chemokine receptor type 4* (*CXCR4*), *Erb-B2 receptor tyrosine kinase 2* (*ERBB2*), *lymphocyte antigen 96* (*LY96*), *myeloid differentiation primary response 88* (*MYD88*), *onco-suppressor Tp53*, *nuclear factor kappa-light-chain-enhancer of activated B cells* (*NF-KB/p65*), *phosphatase and tensin homolog* (*PTEN*), *DNA repair RAD51*, and *transforming growth factor β* (*TGFB*) were evaluated in CF33 cell line at different passages (37th, 39th, and 42nd). These genes were selected from the literature based on their expression patterns in dog mammary cancer [3] and involvement in Salmonella spp. infection [13,22]. Primer sequences are listed in Table 2 [20,23].

Extraction of total RNA was conducted from 1 × 10^6^ cells with the RNeasy Mini Kit (Qiagen s.r.l., Milan, Italy), following the manufacturer’s instructions. RNA concentration and purity were evaluated by BioPhotometer (Eppendorf, Milan, Italy) and Qubit 3.0 Fluorometer (ThermoFisher Scientific, Milan, Italy). Reverse transcription was performed using the OneScript^®^cDNA Syntesis Kit with 250 ng of RNA and including RT-negative controls. Real-time qPCR amplification was carried out on a CFX96™ Real-Time System using SYBR Green chemistry (Applied Biological Materials Inc. Richmond, BC, Canada). Reactions contained 1 × SYBR Green Mix, 0.2 µM of each primer and 100 ng of cDNA in a final volume of 20 µL. A negative control was included in every run. Previously reported thermal profiles [24] were used. To assess basal gene expression, the PCR cutoff was set at Cq 38 (positive samples showed Cq values < 38). The relative gene expression was calculated using the formula: 2^-∆∆Cq^,(1)

### 2.4. Response to the Infective Stressor

The evaluation of CF33 response to *S. Typhimurium* (*ST*) or S. 4,[5],12:i: - (*S. Typhimurium Monofasic STM)* was assessed using the model described in our previous study [22]. Briefly, wildtype *ST* was grown overnight at 37 °C in LB (LB Broth, Miller Luria–Bertani), and then inoculated into fresh medium to obtain mid-log phase culture. *ST* or *STM* was re-suspended at 10^8^ CFU/mL, and 1 mL of this bacterial suspension (MOI 100 CFU/cells) was applied to CF33 cells for 1 h at 37 °C in 5% CO_2_. Cells exposed to *ST* or *STM* were used to evaluate: 1) invasiveness caused by *ST* or *STM* stimulation and 2) immunomodulation.

#### 2.4.1. Bacterial Invasion Assessment

Briefly, after *ST* or *STM* exposure, only washing was applied to cells, which were then treated with 300 μg/mL of colistin sulphate (Microbiol & C. s.n.c., Cagliari, Italy) and incubated in BME at 37 °C in 5% CO_2_ for 2 h to for the complete removal of extracellular bacteria. Preliminary assays had been conducted to verify the lack of toxic side-effects in CF33. Cells were lysed adding 1% Triton X-100 (Sigma, Saint Louis, MO, USA) in PBS. Afterwards, PBS was added to each well, obtaining a cell suspension which was serially diluted and seeded on XLD (Sigma, Saint Louis, MO, USA) and incubated at 37 °C for 24–48 h. After incubation the presence of colonies was assessed. The experiment was performed three times [22].

#### 2.4.2. Modulation of Innate Immune Response

After *ST* or *STM* exposure, the CF33 samples were submitted to three washes with medium and incubated at 37 °C in 5% CO_2_ for a further 3 h with fresh completed medium. The experiment was repeated three times. The negative control was represented by cells treated only with medium. The expression of *IL6*, *CXCL8*, *IL18*, *TLR4*, *TLR5*, *CD14*, *CD44*, *CXCR4*, *LY96 MYD88*, *NF-KB/p65*, *TGFB*, and *TP53* was carried out as described in Section 2.3.

### 2.5. ELISA Assay for IL6

*IL*6 release was determined in supernatants from cells untreated and exposed to *ST* or *STM* using an ELISA kit (R&D system, Inc., Minneapolis, MN, USA) as already described [23]. Cytokine concentration was calculated from ten two-fold dilutions of canine recombinant *IL6*. Plates were read at 492 nm with a TECAN Sunrise microplate reader (Tecan Trading AG, Switzerland). Prism 5 (Graph Pad Software) was used for data analysis; the LOQ (limits of quantification) corresponded to 31.3 pg/mL and the assay range corresponded to 31.3–2000 pg/mL.

### 2.6. Sequencing of CXCR4

The presence of mutations in CXCR4 gene was evaluated by direct sequencing of two CF33 passages (48th and 58th). DNA was extracted from 1 × 10^6^ cells or from 20 mg of tissue using a QIAmp DNA Mini kit (Qiagen s.r.l., Milan, Italy). DNA elution was conducted with 100 μL of Tris-EDTA buffer (TE). In order to analyze a fragment of 902 bp, a set of primers was specifically designed on a reference sequence deposited in GeneBank database (DQ182699.1) with Primer3 software version 0.4.0 (Table 3). For the CXCR4 amplification and sequencing, a protocol previously described in [20] was used. In order to obtain a consensus, BioEdit Sequence Alignment Editor version 7.2.5 was used to align forward and reverse sequences. Examples of consensus from CF33 were compared to canine and human reference sequences (DQ182699.1 and AF348491.1, respectively) through ClustalW multiple alignment [25].

### 2.7. Statistical Analyses

All experiments were replicated at least three times. Significance was set at *p* < 0.05. Data were submitted to a Kolmogorov–Smirnov test to check Gaussian distributions. Student’s *t*-tests and a one-way ANOVA test were performed to determine statistical significance. Student’s unpaired or paired two-tailed *t*-test was applied for comparing two normally distributed groups of samples, while the non-parametric Mann-Whitney’s U-test was used in case of non-normally distributed sample groups. A one-way ANOVA, followed by multiple comparison tests, were used to compare >two normally distributed sample groups. Alpha levels for all tests were 0.05%. Prism (GraphPad Software, Inc., version 5, Motulsky, San Diego, CA, USA) software was used for all statistical analysis.

## 3. Results

### 3.1. Selection of Reference Genes

Twelve genes were identified as the most stable reference genes based on the literature (Table 1). In particular, according to NormFinder analysis the S5 gene resulted the most stable reference gene for the normalization of the CF33 gene expression data (Table 4).

### 3.2. Basal Gene Expression Profile

Our comparison of basal genes expression level between different passages of CF33 showed that the cell line did not express some genes under investigation, such as *IL1B*, *IL2*, *IL10*, *IL15*, *IL17A*, *IL27*, and *IFNG* (Table 5). Instead, all samples expressed similar levels of *IL5*, *CXCL8*, *IL16*, *IL18*, *BRCA1*, *CD44*, *LY96*, and *MYD88* genes. Furthermore, the target genes *CD14*, *CXCR4*, *ERBB2*, *IL4*, *IL6*, *IL12B*, *IL23A*, *NF-KB/p65*, *TNFA*, *TLR4*, *TLR5*, *TP53*, and *TGFB1* were expressed only in some of the analyzed samples and in a variable manner (Table 5).

### 3.3. Response to Infective Stressor

To assess CF33 line response to infective stressors, cell–pathogen interaction tests were performed using the *ST*/*STM* model. The experiments carried out led to the following results.

#### 3.3.1. Modulation of Innate Immune Response

CF33 immunomodulation after *ST* or *STM* treatment was evaluated by analyzing expression of the following genes which participate in cell response: *IL6*, *CXCL8*, *IL18*, *TLR4*, *TLR5*, *CD14*, *LY96*, *MYD88*, *TP53*, *NF-KB/p65*, and *TGFB1* [23,26].

*ST* exposure caused a significantly higher expression of *CXCL8* (*p* = 0.0039) and *CD14* (*p* = 0.0060) and a decrease of *MYD88* (*p* = 0.0410), *NF-KB/p65* (*p* = 0.0458), *TP53* (*p* = 0.0120), LY96 (*p* = 0.0310), and *TLR5* (*p* = 0.0022) in comparison to control cells (C: not treated with *ST* or *STM*) (Figure 1).

The exposure to *STM* determines a more marked increase of *CXCL8* and *CD14* expression compared with *ST* treatment and a decrease of *MYD88*, *NFKB/p65*, *TP53*, *LY96*, and *TLR5* (Figure 1).

Interestingly, we found a marked increase of *IL6* expression after the *ST* exposure in comparison with the *STM* (Figure 2).

#### 3.3.2. Bacterial Invasion Assessment

The invasion test showed that *ST* or *STM* had an identical ability to colonize the cell line after 1 h of exposure (4.58 ± 0.18, 4.58 ± 0.28 log10 bacterial/10^6^ cells, respectively; Figure 3).

### 3.4. IL6 ELISA Assay

The ELISA assay for IL6 protein showed an increase in the release of cytokine by CF33 (170 ± 45 pg/mL in cells treated with *ST* and 150 ± 50 pg/mL in cells treated with *STM*) with respect to the control (Figure 4).

### 3.5. CXCR4 Sequencing

The comparison among the sequences obtained from CF33 cells and human and canine CXCR4 reference sequences available in GeneBank showed the absence of mutations in the gene region analyzed (Figure 5).

## 4. Discussion

Human breast cancer (HBC) is the major cause of cancer-associated mortality of women globally [27]. In 2020, around 2.3 million breast cancer cases were diagnosed [28]. Thus, HBC prevention and innovative and effective therapeutic intervention are the main objectives for many researchers. However, due to limitations in the availability of human tissue samples and ethical issues for clinical research in humans, in vivo and in vitro HBC models are strongly needed. Dogs are considered a good model, due to the higher breast cancer frequency in this species and to the more abundant availability of tissue samples [1,12,29,30]. However, currently the 3R guideline for protecting laboratory animals encourages the use of alternative experimental methods, such as cell cultures [3]. In this regard, it is important to highlight that research involving the use of cell lines requires detailed knowledge on the phenotype, proteins, and basal genes expression. Therefore, the first aim of our study was the characterization of the canine mammary carcinoma cell line CF33. Firstly, the basal expression of genes modulating the innate immune response and regulating the cell cycle was assessed. Secondly, we evaluated this cell line’s interaction with two types of infectious stressors, which are currently studied as new anti-cancer therapies [31]. Moreover, we demonstrated that the CF33 cell line at different passages shows the same characteristics of gene expression as canine mammary carcinoma (CMC) [32,33,34,35,36,37].

Concerning the gene expression evaluation, CF33 showed basal expression of the *BCRA1*, *ERB-B2*, and *RAD51* genes involved in cancer development and progression. In particular, a link between the expression of *BRCA1* and *RAD51* and the development of mammary tumors was hypothesized in dogs [34,38]. These data are in accordance with what has been highlighted in mammary cancer [34,38]. In addition, alterations in TP53 expression may favor tumor development [39,40]. The normal concentration of p53 protein in healthy cells is low, and the protein has a short half-life. This is noteworthy considering that this protein is able to halt cell growth, apoptosis, and cellular senescence, and TP53 mutation are associated with HBC and CMC [3,41]. In CF33, we showed TP53 gene expression only in a proportion of the samples; this suggests an abnormal activity of this gene.

The role of *PTEN*, one of the main tumor suppressor genes, in down-regulation or mutation/deletion in breast cancers has been shown in other studies [42]. Herein we report its non-continuous expression; other studies have shown that copy number aberrations of PTEN are associated with negative evolution of CMCs and HBC [42,43]. In the study by Asproni et al. [44], the authors reported that reduction of *PTEN* immunohistochemical expression can lead to the interruption of the inhibition of the PI3K/AKT pathway, resulting in glycolysis activation for ATP production compared to mitochondrial respiration; therefore PTEN immunohistochemical expression is correlated with less aggressive tumors, no lymphatic invasion, and enhanced survival rates. P-AKT expression is correlated with more aggressive subtypes, lymphatic invasion, and a lower survival rate [44,45]. In our study, the *ERBB2* gene also showed a discontinued expression. It encodes an epidermal receptor which is involved in pathways promoting cell growth and differentiation [46]. An association of *ERBB2* expression with metastasis and poor prognosis both in human and dog has been observed in some studies [47,48]; however, its role remains controversial. *TLR4* and *TLR5* expression was then evaluated. These belong to a group of receptors expressed by several cells (such as dendritic cells, neutrophil granulocytes, B lymphocytes, endothelial cells, macrophages, and mucous epithelium cells) that are able to recognize pathogens and microorganisms, setting off the innate immune response [49]. In detail, *TLR5* recognizes flagellin, while *TLR4* lipopolysaccharides are linked to LBPs (LPS binding proteins). *TLR4* acts by connecting to LY96 and forming a complex with the *CD14* membrane protein [50]. *LY96*, *CD14*, *TLR5* and *TLR4* expression in CF33 suggests a possible interaction with gram-negative bacteria [51]. In this study, *CXCL8* was expressed in all samples. This gene encodes for an important chemokine known for its pro-inflammatory activity and its chemotactic role on basophiles, eosinophils, neutrophils, and others immune cells. Moreover, it is a signaling protein derived from macrophages infiltrating the tumor and it can act as prognostic marker [52,53].

Moreover, we demonstrated the basal gene expression of both CXCR4 and CD44 in the CF33 cell line. The first encodes for an exclusive SDF-1 cell receptor expressed by NK, T cells, monocytes, and dendritic cells [37]. This protein regulates the immune response, being involved in B cells’ development and functioning, leukocyte distribution in peripheral tissues, and lymph node organization [54,55]. Moreover, during bacterial infection, it modulates the transfer of neutrophils between lymph nodes. CXCR4 has a fundamental role in tumor development and progression, and in the metastatic spread of mammary cancer [56,57]. Genetic variation within cell lines may bias results and undermine the reproducibility of cancer research [58]. No development of mutations in CF33 by this receptor was shown in this study; this result is very interesting due to the possibility of using this cell line to test in vitro anticancer therapies taking advantage of CXCR4 pathway.

CD44 is an adhesion molecule which is used by many immune cells (lymphocytes and leucocytes) to adhere to extracellular matrix during inflammation [59]. Data obtained in the preliminary phase confirmed that CF33 has the same gene expression of spontaneous mammary cancer described in previous studies [32,33,34,35,36,37].

Finally, we focused our attention on the ability of CF33 cells to respond to *S. Typhimurium* as infectious stressor. This cell line showed an inflammatory response to *ST*, demonstrated by an up-regulation of IL6 and IL8, which are pro-inflammatory cytokines. On the other hand, we demonstrated *ST*’s ability to penetrate the CF33 cells and reduce the cells’ viability. These results are also in line with a study by Chirullo et al. [13] reporting that these bacteria exploit inflammation to penetrate enterocytes and suggesting the possible use in anti-cancer therapy of attenuated *S. Typhimurium* [13].

Overall, these results confirm the CF33 cell line to be useful for oncological studies and for the development of innovative therapeutic approaches involving the use of bacteria against cancer.

## 5. Conclusions

In conclusion, in agreement with the literature, we demonstrated that CF33 at various passages shows the same characteristics of genes expression as CMC. Our work provides the basal gene expression of the CF33 cell line, which is characterized by overexpression of *CXCR4*, *CD44*, *RAD51*, and *LY96* and a discontinuous expression of *PTEN* and *TP53*. Moreover, no mutations appeared in the CXCR4 gene: this result could represent an important information for studies using the CXCR4 pathway as a therapeutic target. Moreover, the CF33 cell line was shown to be able to interact with *Salmonella Typhimurium* (*ST*), used as infectious stressor, indicating that these cells may be useful as an in vitro model for developing innovative therapeutic approaches involving bacteria.

## Figures and Tables

**Figure 1 vetsci-09-00543-f001:**
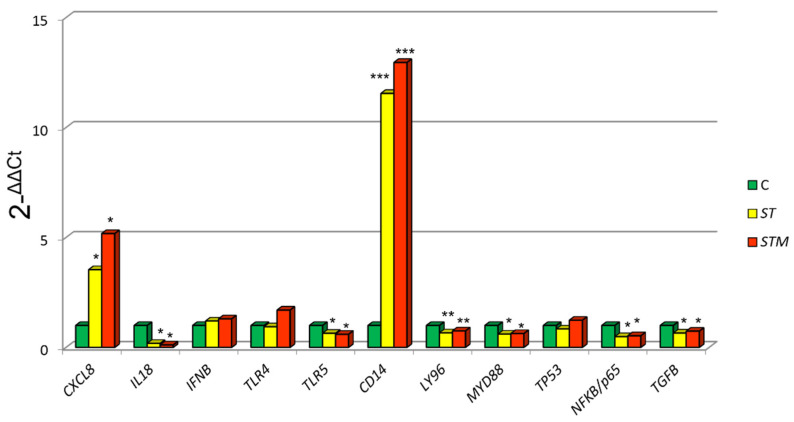
CF33 gene expression modulation after *ST* or *STM* exposure. Data were expressed as 2^-∆∆Cq^ = (Target Cq − S5Cq) − (control Cq − S5 Cq). Significance of expression data was calculated by *t*-test. * *p* < 0.01; ** *p* < 0.001; *** *p* < 0.0001 respect cell control (C: not treated with *ST* or *STM*).

**Figure 2 vetsci-09-00543-f002:**
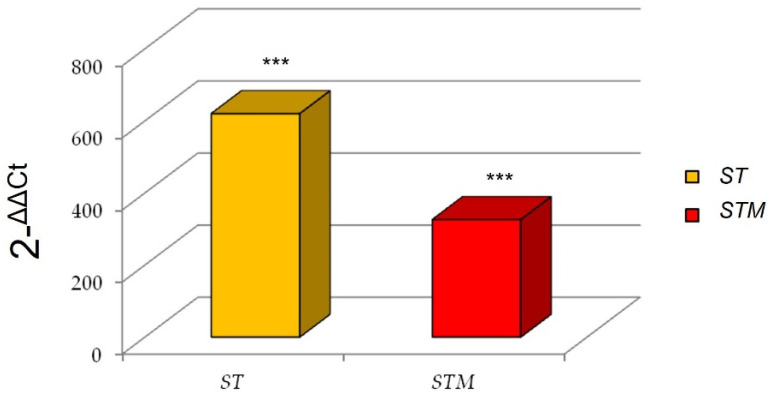
IL6 expression after immunostimulation of CF33 line consequent to *ST* or *STM* exposition. Significance of expression data was calculated by *t*-test. *** *p* < 0.0001 respect to control cells.

**Figure 3 vetsci-09-00543-f003:**
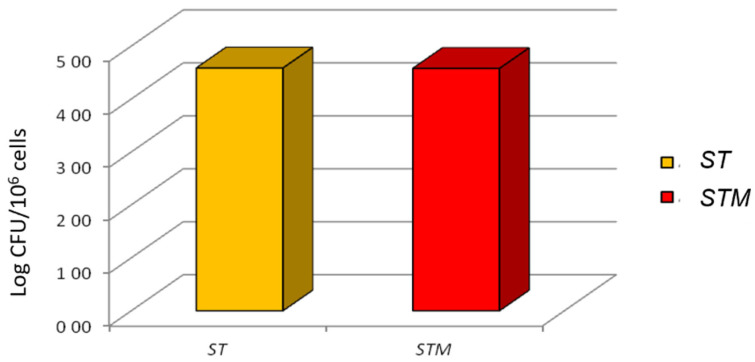
Invasion test: *ST* and *STM*’s ability to invade CF33 cells; the CFUs grown on XLD agar plates are reported on the y-axis.

**Figure 4 vetsci-09-00543-f004:**
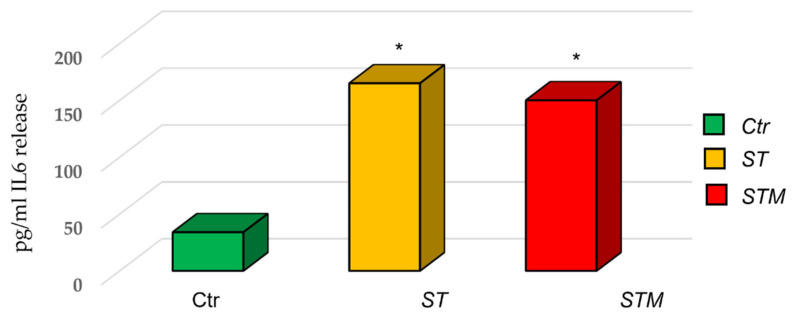
ST and STM’s effect on IL6 release by CF33 cells. Significance of data was calculated by *t*-test. * *p* < 0.05.

**Figure 5 vetsci-09-00543-f005:**
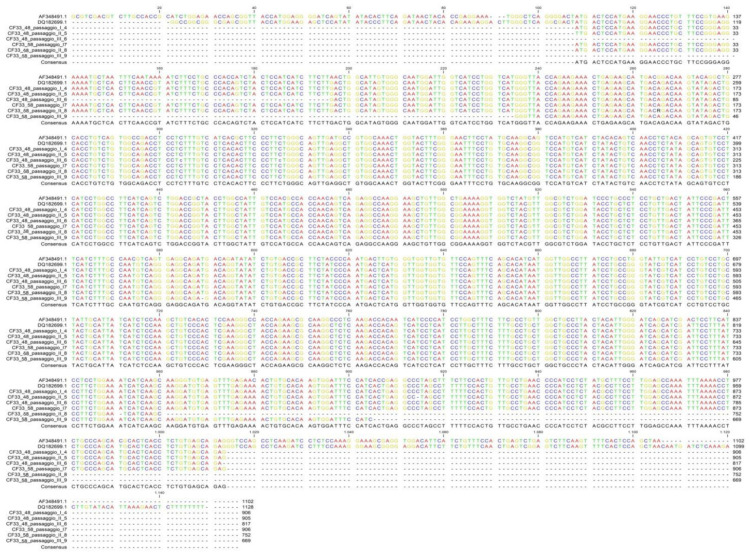
CXCR4 sequencing: alignment of the obtained sequences with CXCR4 human reference (AF348491.1) and CXCR4 canine reference (DQ182699.1) sequences.

**Table 1 vetsci-09-00543-t001:** Canine reference genes: details of primers and amplicons for each evaluated gene.

Gene	Primers	Product Length (bp)	Accession Number
*RPS5*	F: TCACTGGTGAGAACCCCCT R: CCTGATTCACACGGCGTAG	141	XM_533568
*B2M*	F: TCCTCATCCTCCTCGCT R: TTCTCTGCTGGGTGTCG	84	NM_001284479.1
*GUS*B	F: AGACGTTCCAAGTACCCC R: AGGTGTGGTGTAGAGGAGCAC	102	NM_001003191
*GAPDH*	F: CTGGGGCTCACTTGAAAGG R: GGAGGCATTGCTGACAATC	131	NM_001003142.2
*HPRT1*	F: CTGAAGAGCTACTGTAATGACCAGTC R: CTTTTCACCAGCAAGCTTGCAACC	197	NM_001003357.2
*HRNPH1*	F: CTCACTATGATCCACCACG R: TAGCCTCCATAACCTCCAC	150	XM_025446760.3
*BACT*	F: ACGGAGCGTGGCTACAGC′ R: TCCTTGATGTCACGCACGA	61	NM_001195845.3
*SRPR*	F: GCTTCAGGATCTGGACTGC R: GTTCCCTTGGTAGCACTGG	80	XM_03866445.1
*TBP*	F: TCCACAGCCTATCCAGAACA R: CTGCTGCTGTTGTCTCTGCT	66	XM_005627735.4
*RPL13A*	F: GGGGCAGGTCCTGGTGCTCG R: CCAGGTACTTCAACTTGTTTCTGTAG	158	NM_001313766.1
*RPS19*	F: CCTTCCTCAAAAAGTCTGGG R: GCTGTGGAAGCAGCTCGC	124	XM_005616513.4
*SDHA*	F: GGTGGCACTTCTACGACACC R: CCATAATTCTCCAGCTCTACC	112	XM_014110317.3

**Table 2 vetsci-09-00543-t002:** Primer Sets for RT-qPCR evaluation of gene expression in dog.

Gene	Primers	Product Length (bp)	Accession Number
*IL1B*	F: TGCAAAACAGATGCGGATAA R: GTAACTTGCAGTCCACCGATT	64	NM_001037971.1
*IL2*	F: CCTCAACTCCTGCCACAATGT R: TGCGACAAGTACAAGCGTCAGT	71	NM_001003305.2
*IL4*	F: TGCAGAGCTGCTACTGTACTGCGGC R: CATGCTGCTGAGGTTCCTGT	90	NM_001003159.1
*IL5*	F: GCCTATGTTTCTGCCTTTGC R: GGTTCCCATCGCCTATCA	106	NM_001006950.1
*IL6*	F: TCCAGAACAACTATGAGGGTGA R: TCCTGATTCTTTACCTTGCTCTT	100	NM_001003301.1
*CXCL8*	F: TGATTGACAGTGGCCCACATTGTG R: GTCCAGGCACACCTCATTTC	77	NM_001003200.1
*IL10*	F: CGACCCAGACATCAAGAACC R: CACAGGGAAGAAATCGGTGA	101	NM_001003077.1
*IL12B*	F: TGGAGGTCAGCTGGGAATACC R: TGCAAAATGTCAGGGAGAAGTA	69	NM_001003292.1
*IL15*	F: ACTTCCATCCAGTGCTACTT R: CGAGCGAGATAACACCTAAC	271	NM_001197188.1
*IL16*	F: CCAGTCCAAGGGGATTACAG R: TGAGAATGAGCGGTTGTG	100	XM_005618407.4
*IL17A*	F: ACTCCAGAAGGCCCTCAGATTA R: GATTCCAAGGTGAGGTAGATCG	51	NM_001165878.1
*IL18*	F: CTCTCCTGTAAGAACAAAACTATTTCCTT R: GAACACTTCTCTGAAAGAATATGATGTCA	100	NM_001003169.1
*IL23A*	F: ACAGAACGGACAGCATCAGG R: CGCTGCCTGCTTCTCAAATC	101	XM_538231.7
*IL27*	F: TTACTGCTCTCCCTGCTCCT R: TTGAACTCCCTCCGCAACTC	101	XM_038668726.1
*IFNG*	F: CCAGATCATTCAAAGGAGCA R: CGTTCACAGGAATTTGAATCAG	116	NM_001003174.1
*TNFA*	F: CGTCCATTCTTGCCCAAAC R: AGCCCTGAGCCCTTAATTC	94	NM_001003244.4
*TLR4*	F: GCTGGATGGTAAACCGTGGA R: AGCACAGTGGCAGGTACATC	158	NM_001002950.3
*TLR5*	F: CCAGGACCAGACGTTCAGAT R: GCCCAGGAAGATGGTGTCTA	109	NM_001197176.1
*BRCA1*	F: CAGAGAGATACCATGCAAGATAAC R: CTCTTTCTGATGCGTTTTGTTCCG	172	NM_001013416.1
*CD14*	F: GCCGGGCCTCAAGGTACT R: TCGTGCGCAGGAAAAAGC	61	XM_843653.6
*CD44*	F: CAAGGCTTTCAACAGCACCC R: TACGTGTCGTACTGGGAGGT	192	NM_001197022.2
*CXCR4*	F: GCGTCTGGATACCTGCTCTC R: GATACCCGGCAGGATAAGGC	163	DQ182699.1
*ERBB2*	F: CTGAGGGCCGATATACCTTC R: TCACCTCTTGGTTGTTCAGG	114	NM_001003217.3
*LY96*	F: GGGAATACGATTTTCTAAGGGACAA R: CGGTAAAATTCAAACAAAAGAGCTT	92	XM_848045.5
*MYD88*	F: GAGGAGATGGGCTTCGAGTA R: GTTCCACCAACACGTCGTC	160	XM_534223.7
*TP53*	F: CGTTTGGGGTTCCTGCATTC R: CACTACTGTCAGAGCAGCGT	232	NM_001389218.1
*NF-KB/p65*	F: TGTAAAGAAGCGGGACCTGG R: AGAGTTTCGGTTCACTCGGC	250	XM_038424975.1
*PTEN*	F: GTGAAGCTGTACTTCACAA R: CTGGGTCAGAGTCAGTGGTG	136	NM_001003192.1
*RAD51*	F: GGAGAAGGAAAGGCCATGTA R: GGGTCTGGTGGTCTGTGTT	148	NM_001003043.1
*TGFB*	F: CAAGTAGACATTAACGGGTTCAGTTC R: GGTCGGTTCATGCCATGAAT	70	XM_038656896.1

**Table 3 vetsci-09-00543-t003:** Primers for the amplification and sequencing of CXCR4.

Primer		Position	Product Length (bp)	Accession Number
*CXCR4 F*	TCT GTG GCA GAC CTC CTC TT	F 266–285 R 611–630	364	NM_001048026.1
*CXCR4 R*	TGA AAC TGG AAC ACC ACC AA			
*CXCR4 F7*	TGA CTC CAT GAA GGA ACC CTG	F 88–108 R 971–990	902	NM_001048026.1
*CXCR4 R2*	CTG CTC ACA GAG GTG AGT GC			
*CXCR4 Fow 3a*	GTC ATC CTG TCC TGC TAC TG	F 665–684 R 296–314	Sequencing 902 bp	NM_001048026.1
*CXCR4 Rev 3b*	CAA CTG CCC AGA AGG GAA G			

**Table 4 vetsci-09-00543-t004:** Stability of reference genes (RG).

Gene	Stability (p)
*B2M*	0.031
*BACT*	0.027
*RPS5*	0.014
*RPS19*	0.041
*GAPDH*	0.045
*HPRT1*	0.051
*RPL13A*	0.04
*HRNPH1*	0.051
*SRPR*	0.042
*TBP*	0.048
*SDHA*	0.039
*GUSB*	0.044

**Table 5 vetsci-09-00543-t005:** Basal expression of inflammatory and immunomodulatory genes in CF33 cells. Data are expressed as: + all samples were positive; − all samples were negative; ± only some samples were positive. ND not determined.

Gene	Expression	Cq ± SD
*IL1B*	−	ND
*IL2*	−	ND
*IL4*	±	38.3 ± 0.1
*IL5*	+	37.7 ± 0.8
*IL6*	±	38.0 ± 0.9
*CXCL8*	+	35.7 ± 0.8
*IL10*	−	ND
*IL12B*	±	38.7 ± 0.1
*IL15*	−	ND
*IL16*	+	38.7 ± 0.5
*IL17A*	−	ND
*IL18*	+	37.7 ± 0.6
*IL23A*	±	37.3 ± 0.6
*IL27*	−	ND
*IFNG*	−	ND
*TNFA*	±	37.1 ± 1.1
*TLR4*	±	38.2 ± 0.6
*TLR5*	±	38.1 ± 1.0
*BRCA1*	+	32.5 ± 0.8
*CD14*	±	37.8 ± 0.8
*CD44*	+	37.1 ± 1.1
*CXCR4*	±	36.8 ± 0.3
*ERB-B2*	±	38.1 ± 0.6
*LY96*	+	27.1 ± 0.5
*MYD88*	+	37.2 ± 0.7
*TP53*	±	35.8 ± 0.5
*NF-KB/(p65)*	±	38.8 ± 0.7
*PTEN*	±	38.3 ± 1.2
*RAD51*	+	32.5 ± 1.3
*TGFB1*	±	37.1 ± 3.8

## Data Availability

Not applicable.

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
