# Peer review of "Molecular Characterization of CF33 Canine Cell Line and Evaluation of Its Ability to Respond against Infective Stressors in Sight of Anticancer Approaches"

_vetsci, 2022, doi:10.3390/vetsci9100543_

Round 1
Reviewer 1 Report
The authors evaluated the gene expression profile of CF33 cell lines and investigated the interaction of infective stressor such Salmonella Typhimurium. The present work is well structured, methodologically correct, and fluent in reading; therefore, I have only minor suggestions to address.
Lines 118-130: Among the genes of interest, have you thought about evaluating ER and PR which are important in the molecular classification of canine mammary carcinomas?
Line 131: Format the table to fit on a page
Line 158: Please insert “h” to “te”.
Line 191: What test was used to establish the normality? Please add it.
Linea 304: Please explain the OSA acronym.
Linea 304: The bibliographic references are referred to mammary tumors and not to p53 and OSA. Please check the bibliography.
Line 309: Use only the acronym as it has already been explained above.
Line 322-323: The role of ERBB2 in canine mammary tumors is controversial, this should be briefly discussed here. Some authors have found that it does not play a relevant role or even a better prognosis in canine mammary tumors.
Author Response
The authors wish to thank the Reviewer for the time spent in revising their paper. All comments were adressed. Please find detailed answers below.
Reviewer 1
The authors evaluated the gene expression profile of CF33 cell lines and investigated the interaction of infective stressor such Salmonella Typhimurium. The present work is well structured, methodologically correct, and fluent in reading; therefore, I have only minor suggestions to address.
Lines 118-130: Among the genes of interest, have you thought about evaluating ER and PR which are important in the molecular classification of canine mammary carcinomas?
We thank the reviewer for this question. We agree with the importance of ER and PR in the classification of canine breast tumors, however in this work we have considered others aspect of mammary cancer.
Line 131: Format the table to fit on a page
Done
Line 158: Please insert “h” to “te”.
Done
Line 191: What test was used to establish the normality? Please add it.
Data were submitted to a Kolmogorov-Smirnov test to check Gaussian distributions. This information was added in the text.
Linea 304: Please explain the OSA acronym.
The sentence was slightly changed.
Linea 304: The bibliographic references are referred to mammary tumors and not to p53 and OSA. Please check the bibliography.
There was a mistake, we have now inserted the correct tumor
Line 309: Use only the acronym as it has already been explained above.
Done
Line 322-323: The role of ERBB2 in canine mammary tumors is controversial, this should be briefly discussed here. Some authors have found that it does not play a relevant role or even a better prognosis in canine mammary tumors.
Done
Reviewer 2 Report
The present study aims to evaluate the interaction of CF33 and Salmonella Typhimurium as ineffective stressor, showing as well their utility as an in vitro model for evaluating therapeutic approaches by means of bacteria. The manuscript is interesting since it demonstrating the utility of specific cell lines as in vitro therapies for canine mammary cancer.
The manuscript highlights an interesting topic and shows nice results from both points of view, the results and the possibility of an alternative to laboratory animals. However, it needs some English revision, both grammatical and orthographic. Please, submit it to a language reviewer before re-submitting it to the journal.
L100. Change microgolbulin by microglobulin.
L128. Change 42th by 42nd
L269. Remove the comma.
L271. The fragment “likely related to the increase of IL expression” should be moved to the discussion chapter, since it’s not results.
L295-299. This is not discussion but conclusions.
The whole discussion chapter needs a deep revision of language. Many sentences are not properly written, making its understanding difficult.
Why did the authors used passages 37th, 39th and 42nd specifically?
Author Response
The authors wish to thank the Reviewer for the time spent in revising their paper. All comments were adressed. Please find detailed answers below.
Reviewer 2
The present study aims to evaluate the interaction of CF33 and Salmonella Typhimurium as ineffective stressor, showing as well their utility as an in vitro model for evaluating therapeutic approaches by means of bacteria. The manuscript is interesting since it demonstrating the utility of specific cell lines as in vitro therapies for canine mammary cancer.
The manuscript highlights an interesting topic and shows nice results from both points of view, the results and the possibility of an alternative to laboratory animals. However, it needs some English revision, both grammatical and orthographic. Please, submit it to a language reviewer before re-submitting it to the journal.
Editing of English was done by a mother-toungue
L100. Change microgolbulin by microglobulin.
Done
L128. Change 42th by 42nd
Done
L269. Remove the comma.
Done
L271. The fragment “likely related to the increase of IL expression” should be moved to the discussion chapter, since it’s not results.
Done
L295-299. This is not discussion but conclusions.
Modified
The whole discussion chapter needs a deep revision of language. Many sentences are not properly written, making its understanding difficult.
The text was revised by a mother-tongue.
Why did the authors used passages 37th, 39th and 42nd specifically?
We decided to use a number of passages that can reproduce the routine of labs